# Metal-Free Phosphination and Continued Functionalization of Pyridine: A Theoretical Study

**DOI:** 10.3390/molecules27175694

**Published:** 2022-09-03

**Authors:** Pan Du, Yuhao Yin, Dai Shi, Kexin Mao, Qianyuan Yu, Jiyang Zhao

**Affiliations:** 1School of Life Science and Chemistry, Jiangsu Second Normal University, Nanjing 210013, China; 2School of Environmental Science, Nanjing Xiaozhuang University, Nanjing 211171, China

**Keywords:** phosphination, metal free, C–H fluoroalkylation, mechanism, DFT

## Abstract

This study investigates the mechanism of metal-free pyridine phosphination with P(OEt)_3_, PPh_3_, and PAr_2_CF_3_ using density functional theory calculations. The results show that the reaction mechanism and rate-determining step vary depending on the phosphine and additive used. For example, phosphination of pyridine with P(OEt)_3_ occurs in five stages, and ethyl abstraction is the rate-determining step. Meanwhile, 2-Ph-pyridine phosphination with PPh_3_ is a four-step reaction with proton abstraction as the rate-limiting step. Energy decomposition analysis of the transition states reveals that steric hindrance in the phosphine molecule plays a key role in the site-selective formation of the phosphonium salt. The mechanism of 2-Ph-pyridine phosphination with PAr_2_CF_3_ is similar to that with PPh_3_, and analyses of the effects of substituents show that electron-withdrawing groups decreased the nucleophilicity of the phosphine, whereas aryl electron-donating groups increased it. Finally, TfO^−^ plays an important role in the C–H fluoroalkylation of pyridine, as it brings weak interactions.

## 1. Introduction

Organophosphorous compounds have broad applications in the fields of organic synthesis, ligand chemistry, medicinal chemistry, agrochemistry, and materials [1,2,3]. In light of their fundamental importance, many methods of C-P bond formation have been developed [4,5,6]. Among these methods, transition-metal-catalyzed cross-coupling reactions, including the Pd- [7], Cu- [8], Zn- [9], Ni- [10], Mn- [11], Ag- [12], and Rh-catalyzed [13] phosphination reactions of various aryl partners with phosphine reagents, have received great attention [14]. In particular, researchers have focused on developing a mild, efficient, and environmentally benign method of C–P bond formation [15,16,17].

In 1979, the Akiba group succeeded in synthesizing dimethyl phosphonates by reacting quinoline or isoquinoline with acyl chlorides then adding trimethyl phosphite (Figure 1a) [18]. Since then, many improved methods have been developed. For example, Ander et al. reported the synthesis of PO(OR)_2_- and PR_3_-substituted N-heteroaromatic rings by reaction of the N-heteroaromatic ring with phosphorous compounds in the presence of Tf_2_O and amine (Figure 1b,c) [19,20,21,22,23]. Although the formation of metal-free C–P bonds has been investigated for many decades, little progress had been made in this area until McNally and co-workers finally succeeded in developing a general approach to form C–O, C–S, C–C, C–D/T, and C–N bonds by transforming pyridines to phosphonium salts then reacting them with nucleophiles (Figure 1d) [24,25,26,27,28,29,30,31,32]. The groups of Harutyunyan and Jumde were able to produce pyridyl-ether by directly functionalizing the pyridine ring according to the methodology proposed by McNally and co-workers [33]. The metal-free Sandmeyer-type phosphorylation of aryl amines and electrophilic phosphinative cyclization of alkynes were used to synthesize aryl phosphonates [34] and various phosphine derivatives [35]. Finally, Stephan et al. used Frustrated Lewis Pairs to form compounds with C-P bonds [36].

Despite their importance for the construction of phosphorous compounds and phosphonium salts, which can be used as a late-stage functionalization tool, the mechanisms underlying the formation of metal-free C–P bonds are not well understood. McNally et al. proposed a possible reaction pathway for the reaction illustrated in Figure 1c, as shown in Figure 1e [27]. According to this mechanism, the reaction is initiated by the nucleophilic attack of pyridine on Tf_2_O (Ι) to form a pyridinium triflyl salt. The addition of phosphine to this salt results in the formation of a C–P bond (ΙΙ). In the last step, the NEt_3_ base abstracts an H-atom from the adduct to restore aromaticity (ΙΙΙ). Although this mechanism explains how the C–P bond is formed, it does not account for the factor controlling the site-selective formation of the phosphonium salt. In reaction Figure 1b, NaI is involved in the generation of the C–P bond, and the product is neutral. Meanwhile, in reaction Figure 1c, the product is an ion pair. The role of NaI and the reason behind the difference in product nature remain unclear. In this study, we investigate the detailed mechanisms of pyridine reaction with P(OEt)_3_, PPh_3_, and PAr_2_CXF_2_ phosphines, and we assess the effect of the phosphine on the formation of metal-free C–P bonds.

## 2. Results and Discussion

In this section, we discuss the possible mechanisms of pyridine phosphination with P(OEt)_3_, PPh_3_, and PAr_2_CXF_2_. The site-selectivity of the reaction is also analyzed.

### 2.1. Mechanism of Pyridine Phosphination with P(OEt)_3_

The computed Gibbs free energy profiles corresponding to the phosphination of pyridine with P(OEt)_3_ in acetonitrile solvent (Equation (1)) are presented in Figure 1 and Figure 2. The optimized geometries of all stationary points along the reaction pathway are displayed in Appendix A of the Appendix A.

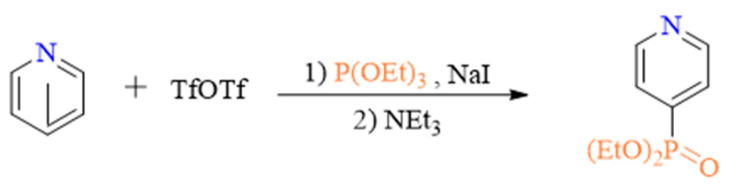
(1)

#### 2.1.1. Activation of Pyridine and Nucleophilic Addition

Based on our calculations, the reaction is initiated by the nucleophilic attack of pyridine on Tf_2_O (S_N_2 reaction) to form *N*-(trifluoromethylsulfonyl) pyridium triflate **2** and TfO^−^ via the **TS_1/2_** transition state [37]. The free energy barrier of this reaction step is only 1.2 kcal/mol in acetonitrile (relative to pyridine + Tf_2_O), which agrees well with the expected barrier value, considering the experimental conditions (−78 °C to room temperature). The S–O(OTf) bond in **TS_1/2_** is elongated to 1.73 Å, whereas the N–S bond distance is decreased to 2.56 Å. The natural population analysis (NPA) charge of the C4 atom of the pyridine moiety increases from −0.159 in free pyridine to −0.06 in intermediate **2**, which indicates that this atom becomes more electrophilic along the reaction pathway. Therefore, Tf_2_O facilitates the nucleophilic addition reaction by strengthening the electrophilicity of pyridine.

The second reaction step involves the addition of the phosphine P(OEt)_3_ nucleophile to intermediate **2** via the **TS_3/4_** transition state, yielding *N*-trifluoromethylsulfonyldihydropyridine **4**. This step has a free energy barrier of 11.1 kcal/mol in acetonitrile (relative to **3**), which is relatively low, and the C4–P bond distance in **TS_3/4_** is 2.37 Å. The approach of P(OEt)_3_ distorts the pyridine ring, and in the dearomatized intermediate **4**, this ring has a typical dihydropyridine structure.

#### 2.1.2. Rearomatization of Dihydropyridine

The rearomatization of dihydropyridine (intermediate **4**) occurs via two possible reaction pathways. The first pathway involves the NaI-mediated migration of an ethyl group, followed by proton abstraction by NEt_3_ (path A in Figure 2). Meanwhile, in the second pathway, abstraction occurs first, then migration (path B in Figure 3).

As shown in Figure 2, path A of dihydropyridine rearomatization is mediated by NaI, which can directly extract the ethyl group in **4** via **TS_5a_**. However, the free energy barrier of this step is high (40.3 kcal/mol). Alternatively, the TfO^−^ ion in **4** may be exchanged with the I^−^ ion of NaI, which promotes the migration of the ethyl group. The iodide ion may attack the ethyl groups from the frontside via **TS_6a_** or from the backside via **TS_7/8_**, with free energy barriers of 35.9 and 20.0 kcal/mol, respectively. The attack of I^−^ on the ethyl groups is an S_N_2 type reaction, where the iodide ion acts as a nucleophile. Considering that the attack of I^−^ from the backside yields higher orbital overlap than the attack from the frontside (Figure 4), **TS_7/8_** is more favorable than **TS_5a_** or **TS_6a_**.

Subsequently, the product of the S_N_2 reaction, 4-Phosphonato substituted pyridine **9**, reacts with NEt_3_ to generate intermediate **11** via a proton transfer transition state (**TS_10/11_**). With a free energy barrier of 13.0 kcal/mol only (relative to **9**), this reaction step occurs readily. Intermediate **11** undergoes isomerization to a more stable isomer **12**, in which the ammonium cation is close to the P=O group. Then, the N-S bond in **12** breaks to form intermediate **13** via **TS_12/13_**, which lies at 6.0 kcal/mol above **9**. The final product **15** is obtained upon the dissociation of the Et_3_HN^+^···CF_3_SO_2_^−^ ion pair **14** from **13**.

In path B, proton abstraction precedes the migration of the ethyl group. The free energy profile of this pathway is shown in Figure 3, and the optimized geometries of all implicated species are given in Appendix A. First, the NEt_3_ amine abstracts a proton from the C4 of the dihydropyridine intermediate **4** via **TS_16/17_** to generate intermediate **17**. The free energy barrier of this process is 8.9 kcal/mol (relative to **4**), which is relatively low. Then, intermediate **17** isomerizes to a more stable isomer **18**, in which the ammonium cation is close to the Tf group. Subsequently, the N-S bond in **18** breaks, resulting in the formation of intermediate **19** via **TS_18/19_**, which lies at 4.1 kcal/mol above **18**. Finally, the iodide ion of NaI abstracts the ethyl group of **19** via an S_N_2 reaction, whose free energy barrier is only 17.8 kcal/mol (**TS_21/22_**). This indicates that the I^−^-mediated migration of the ethyl group is a facile process. In both paths, A and B, ethyl migration is the rate-demining step. The free energy barrier of this step in path B is 17.8 kcal/mol (**TS_21/22_**,), which is comparable to that in path A (**TS_7/8_**, 20.0 kcal/mol). This reveals that the reaction can also occur when the order of the addition of NaI and NEt_3_ in the experiment is reversed.

Overall, our calculations suggest that path A (i.e., the activation of pyridine and nucleophilic addition of P(OEt)_3_, followed by the migration of the ethyl group and rearomatization of dihydropyridine) is energetically feasible, with an overall energy change of −83.0 kcal/mol. The rate-limiting step is the nucleophilic addition of P(OEt)_3_ via **TS_7/8_**, which has a free energy barrier of 20.0 kcal/mol. Although the order of the elementary steps is reversed in path B, this path can also occur under experimental conditions.

#### 2.1.3. Origin of Site-Selectivity

Based on the experiments conducted in a previous study, the reaction of pyridine with P(OEt)_3_ results in the nearly exclusive formation of C4-phosphonates (C4-phosphonate:C2-phosphonate = 95:5) [22]. The implicated mechanism of the formation of C2-phosphonate involves nucleophilic addition (**o-TS_3/4_**, 13.7 kcal/mol) and the migration of the ethyl group (**o-TS_7/8_**, 21.9 kcal/mol), as illustrated in Figure 5. The rate-determining step is ethyl group abstraction by I^−^ (**o-TS_7/8_**). Notably, C2-phosphonates may be formed via the same mechanism as C4-phosphonates; however, the free energy barriers of the key reaction steps are lower in the case of C4-phosphonate formation, as shown in Table 1. Moreover, the energy of the C4-phosphonate product is lower than that of the C2 counterpart. These results indicate that compared to C2-phosphonate formation, the production of C4-phosphonates is more favored both dynamically (**TS_3/4_** and **TS_7/8_**) and thermodynamically (**9**).

The C2-phosphonate **o-9** undergoes proton abstraction by NEt_3_ via the **o-TS_10/11_** transition state. The related free energy barrier is 21.9 kcal/mol, compared with 13.1 kcal/mol for proton abstraction from the C4-phosphonate (**TS_10/11_**). Considering the relatively high barrier of **o-9** rearomatization, this is the main product of the reaction.

To determine the origin of product selectivity, activation strain model (ASM) [38,39,40] analyses of the **TS_7/8_** and **o-TS_7/8_** transition states were conducted. These transition states may be divided into two parts, one involving phosphorane and I^−^, and the other involving pyridine and the Tf group. The energy of the interaction between the two parts in **TS_7/8_** and **o-TS_7/8_** is −106.9 and −113.5 kcal/mol, respectively. The larger interaction energy of **o-TS_7/8_** compared to **TS_7/8_** is attributed to the presence of a strong π bond in the former. As for the strain energies of the phosphorane and pyridine moieties, they are larger in **o-TS_7/8_** than in **TS_7/8_** (Table 2), which indicates that the steric hindrance in the former is greater than that in the latter. Considering that the interaction energy and strain energy differences between **TS_7/8_** and **o-TS_7/8_** are −6.6 and 10.8 kcal/mol, respectively, the free energy of **o-TS_7/8_** is larger than that of **TS_7/8_**, and the para-substituted phosphonate is the main product. In summary, the product selectivity is determined by the steric hindrance. The phosphination of pyridine with P(O*i*Pr)_3_ to afford C2- and C4-phosphonates was also explored, and the results confirm that steric hindrance plays a key role in site-selectivity. The related free energy profiles and optimized structures are presented in Appendix A.

### 2.2. Phosphination of Pyridine with PPh_3_

#### 2.2.1. Mechanism of Pyridine Phosphination with PPh_3_


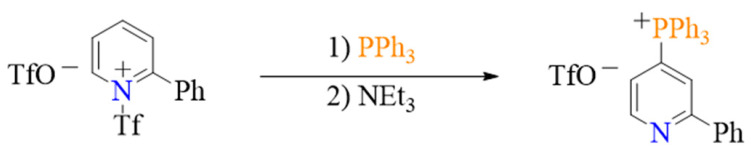
(2)

As there is no NaI additive, the process of 2-Ph-pyridine phosphination with PPh_3_ is simpler than that with P(OEt)_3_ (Equation (2)). The free energy profile of this process is shown in Figure 6, and the geometries of all implicated species are illustrated in Appendix A.

The first and second steps of phosphination with PPh_3_ are similar to those of phosphination with P(OEt)_3_, and their free energy barriers are 7.7 (**TS_23/24_**) and 7.8 kcal/mol (**TS_25/26_**), respectively. Following these steps, the NEt_3_ amine abstracts a proton from C4 of the dihydropyridine intermediate **26** via the **TS_27/28_** transition state to generate intermediate **28**. The free energy barrier of this step is 20.9 kcal/mol (relative to **26**). Subsequently, **28** isomerizes produce intermediate **29** and the S–N bond in this intermediate is broken to give the 4-phosphonato substituted pyridine **31** and an ammonium salt **14**. Considering that the aromaticity of pyridine is restored by breaking the S–N bond in **29**, the free energy barrier of this step is very small (0.4 kcal/mol). Therefore, proton abstraction (**TS_27/28_**, 20.9 kcal/mol) is the rate-limiting step in the mechanism of 2-Ph-pyridine phosphination with PPh_3_. The same step in the mechanism of pyridine phosphination with P(OEt)_3_ has a smaller barrier (**TS_10/11_,** 13.0 kcal/mol) due to the relatively low steric hindrance imposed by P(OEt)_3_, whose volume is smaller than PPh_3_.

#### 2.2.2. Origin of Site-Selectivity

The experiments conducted in previous studies show that the reaction of 2-Ph-pyridine phosphination with PPh_3_ is highly selective and favors the formation of the para-substituted product [19,22]. To investigate the reason behind the site-selectivity of this reaction, calculations of 2-Ph-pyridine phosphination at the ortho position were performed. The obtained free energy profile is shown in Figure 7, and the geometries of all implicated species are shown in Appendix A. The nucleophilic attack of PPh_3_ on the pyridinium salt **24** at the ortho position (**TS_24/32_**) has a free energy barrier of 7.0 kcal/mol, and the barrier of the subsequent proton abstraction by NEt_3_ (**TS_32/33_**) is 22.7 kcal/mol. The latter is higher than the energy barrier of the analogous reaction, leading to the formation of the para-substituted product (**TS_27/28_**, 20.9 kcal/mol). Based on ASM [38,39,40] analysis, the difference in the strain energies of the phosphonate moieties in **TS_27/28_** and **TS_32/33_** are bigger than that of the pyridine and NEt_3_ moieties (Table 3). This indicates that the difference between the energies of the two transition states is mainly attributed to the phosphonate moiety. Specifically, the steric hindrance at the ortho position of pyridine renders this moiety more distorted. Therefore, the phosphination of pyridine at ortho position is less favorable than that at para position, which agrees well with the available experimental data.

Overall, the results indicate that the phosphination of pyridine with PPh_3_ may proceed via four successive elementary steps including activation of pyridine, nucleophilic addition, proton abstraction, and S–N bond breaking. The large volume of PPh_3_ raises the free energy barrier of proton abstraction. Nevertheless, the reaction is favored by its high exothermicity (−72.7 kal/mol). The site-selectivity is thus attributed to the unfavorable steric hindrance at the ortho position of pyridine.

### 2.3. Phosphination of 2-Ph-Pyridine with Diarylfluoroalkylphosphines (PAr_2_CF_2_X)

Paton and Mcnally previously reported the phosphination of 2-Ph-pyridine with diarylfluoroalkylphosphines (Figure 2) [41]. They compared five different phosphines and found that the yield of the product correlates with the donating capacity of the phosphine’s aryl substituents.

Herein, the mechanism of 2-Ph-pyridine phosphination with P1–P5 was studied and the obtained results demonstrate that unlike the phosphination reaction with PPh_3_, the rate-determining step is the nucleophilic addition of phosphines. Compared to PPh_3_ (proton affinity = 159.4 kcal/mol), the proton affinities of P1–P5 are smaller, as shown in Table 4. This suggests that the CF_3_ and CF_2_H electron-withdrawing substituents decrease the nucleophilicity of the diarylfluoroalkylphosphines, thereby increasing the energy barrier of nucleophilic addition. As a result, this reaction step becomes the rate-determining step.

Unlike the electron-withdrawing substituents (CF_3_ or CF_2_H) of P1–P5, the aryl electron-donating substituents (OMe, NMe_2_, and *N*-pyrrolidinyl) promote the nucleophilicity of the phosphines, as per the proton affinity values listed in Table 4. This means that they decrease the barrier of the nucleophilic addition elementary step. Consequently, when the aryl substituent in the diarylfluoroalkylphosphine is an electron-donating group, the nucleophilic addition of this phosphine to 2-Ph-pyridine is facile, and the product yield of the reaction is high. The opposing effects of electron-withdrawing and electron-donating substituents on the rate of 2-Ph-pyridine phosphination with diarylfluoroalkylphosphines agree well with the experimentally observed reactivities of different phosphines [41].

### 2.4. C-H Fluoroalkylation Reaction of 2-Ph-Pyridine

As mentioned in the introduction, the phosphonium ion can be used as a functional handle to form other chemical bonds. For example, Paton and Mcnally used phosphonium salts to form fluoroalkyl pyridine, and they studied this reaction both, experimentally and theoretically (DFT calculations) [41]. Based on the obtained results, the authors proposed that the reaction process involves water addition and ligand coupling, as shown in Figure 3, and that the key transition state is the coupling of CF_3_–PyH^+^ (**CF_3_-PyH^+^-TS**). In the presence of TfO^−^, the free energy barrier of this transition state is reduced from 25.8 to 16.2 kcal/mol, which indicates that this anion plays an important role in the CF_3_-PyH^+^ coupling reaction. Herein, the interactions in the **CF_3_-PyH^+^+TfO-TS** transition state were analyzed, and as shown in Figure 8, there are two types of weak interactions: hydrogen bonding between TfO^−^ and OH, and dispersion effects between TfO^−^ and the pyridinium ring. These interactions promote ligand coupling.

## 3. Computational Methods

The Gaussian16 software package [42] was used to perform all DFT calculations according to the self-consistent reaction field (SCRF) method and the IEFPCM solvation model [43], with acetonitrile as solvent. The geometries of all minima and transition states were optimized and the harmonic frequencies calculated at the M06-2X/6-311G** level of theory in solution [44,45,46]. Meanwhile, single-point energy calculations of the minimum energy conformers (i.e., optimum geometries) were performed at the M06-2X/6-311++G** level of theory in solution. The 3D-optimized structures were visualized using the CYLview program [47], and the calculated frequency values were used to correct the free energy at 298.15 K and 1 atm. Based on the “the theory of free volume” [48,49,50,51,52,53], a correction factor of −2.55 (or 2.55) kcal mol^−1^ was added to the free energy values to account for the effect of the ideal gas phase model in overestimating the contribution of entropy. Meanwhile, the Gibbs energies were corrected to standard state 1 M [54]. The proton affinity were defined as the negative of the enthalpy change for the reaction (P + H^+^ → PH^+^). The important transition states were confirmed using intrinsic reaction coordinate (IRC) analysis [55,56], and the partial atomic charges were allocated based on natural bond orbital (NBO) analyses, which were performed at the M06-2X/6-311G** level [57,58,59]. To study the electronic structure changes induced by the S_N_2 reactions along the reaction pathway, intrinsic bond orbital (IBO) analyses were performed [60]. The weak interactions in the transition state were analyzed using the Multiwfn program [61,62].

## 4. Conclusions

This study uses DFT calculations to elucidate the detailed mechanism of pyridine phosphination with three different phosphines: P(OEt)_3_, PPh_3_, and PAr_2_CF_3_. As shown in Figure 4, the reactions are initiated by pyridine activation and nucleophilic addition of phosphine. In the case of phosphination with P(OEt)_3_, the subsequent steps are NaI-mediated ethyl migration and rearomatization of dihydropyridine. Meanwhile, in the case of phosphination with PPh_3_ or PAr_2_CF_3_, the rearomatization of dihydropyridine by NEt_3_ occurs first, followed by ligand coupling of the 4-phosphonato substituted pyridine intermediate to give trifluoromethylated pyridine. Considering that the proton affinity of PAr_2_CF_3_ is smaller than the affinities of P(OEt)_3_ and PPh_3_, the rate-determining step of pyridine phosphination with PAr_2_CF_3_ is nucleophilic addition, whereas that of phosphination with P(OEt)_3_ or PPh_3_ is ethyl migration and proton transfer.

The steric hindrance of phosphine determines the site-selectivity of the phosphination reaction with P(OEt)_3_ or PPh_3_. In the case of PAr_2_CF_3_, the effect of the aryl electron-donating substituents increasing the proton affinity and reducing the reaction barrier is greater than the opposing effect of the electron-withdrawing substituents, which facilitates the nucleophilic addition of the phosphine. These electronic and steric effects on the rate of the reaction further support our proposed mechanism. Finally, TfO^−^ plays an important role in the C–H fluoroalkylation of 2-Ph-pyridine, as it induces hydrogen bonding interactions and dispersion effects in the ligand coupling transition state.

## Data Availability

Not applicable.

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
