# Peer review of "Metal-Free Phosphination and Continued Functionalization of Pyridine: A Theoretical Study"

_molecules, 2022, doi:10.3390/molecules27175694_

Round 1
Reviewer 1 Report
This article investigated metal-free pyridine phosphination by employing the density functional theory (DFT) calculations. Although I personally do not recommend the out-of-date Pople's split-valence basis sets. The triple-zeta basis sets the author adopted are still reliable. The post-analysis sections are written in high quality. All reaction schemes, molecular structures, and isosurfaces are presented with high quality. This article can be published after minor revision. I suggest the author to pay attention to the following points.
1. The author has mentioned NBO charges (Line 135). However, there is no NBO charge strictly speaking. NBO is the name of the NBO package, or the assembly of a series of analyses based on Natural Bond Orbital (NBO) analysis, including NPA, NAO, NHO, NBO, NLMO, NRT, NBBP, and NEDA. The first step is to generate natural atomic orbitals (NAO). The following analysis is based on the NAO density matrix and the population analysis based on these NAO is available. The atomic charges are obtained from the population analysis of the natural atomic orbitals. Therefore, I strongly suggest addressing them as natural population analysis (NPA) charges.
2. I suggest the author add the definition of proton affinity to the Method section. What is the unit for the proton affinity in Table 4? Does the author use the experimental energy for proton?
3. I understand that M06-2X functional has included the dispersion effect. The semi-empirical DFT-D3 dispersion correction is not necessary for this study. I still recommend the author to use M06-2X accompanied by D3 correction in the future to have a better description of other effects, such as π-π stacking.
Author Response
Many thanks for your good suggestions. We revised the manuscript and improved its quality. Our changes and responses are listed below:
- Comment (The author has mentioned NBO charges (Line 135). However, there is no NBO charge strictly speaking. NBO is the name of the NBO package, or the assembly of a series of analyses based on Natural Bond Orbital (NBO) analysis, including NPA, NAO, NHO, NBO, NLMO, NRT, NBBP, and NEDA. The first step is to generate natural atomic orbitals (NAO). The following analysis is based on the NAO density matrix and the population analysis based on these NAO is available. The atomic charges are obtained from the population analysis of the natural atomic orbitals. Therefore, I strongly suggest addressing them as natural population analysis (NPA) charges.)
Response: NBO charge is changed to natural population analysis (NPA) charge in the revised manuscript.
- Comment (I suggest the author add the definition of proton affinity to the Method section. What is the unit for the proton affinity in Table 4? Does the author use the experimental energy for proton?)
Response: The definition of proton affinity is added in the Computational Methods and its unit is kcal/mol. The unit is added in Table 4. We don’t use the experimental energy for proton in our study.
- Comment (I understand that M06-2X functional has included the dispersion effect. The semi-empirical DFT-D3 dispersion correction is not necessary for this study. I still recommend the author to use M06-2X accompanied by D3 correction in the future to have a better description of other effects, such as π-π stacking.)
Response: The semi-empirical DFT-D3 dispersion correction and related discussions are deleted in the revised manuscript.
Reviewer 2 Report
In this article, the authors very clearly discussed the importance of DFT calculation to understand as well as predict, detailed mechanism of pyridine phosphination with three different phosphines, P(OEt)3, PPh3, and PAr2CF3. Description of electronic structures of the molecules can be obtained from DFT calculations which leads to provide information about the electronic properties of the molecules.
The paper is clear and concise and I have some comments on this work presentation.
Can authors comment on scheme 4 in conclusion part. After nucleophilic addition of the phosphine, initially two double bonds should be in pyridine ring.
Author Response
Many thanks for your good suggestions. We revised the manuscript and improved its quality. Our changes and responses are listed below:
1 Comment (Can authors comment on scheme 4 in conclusion part. After nucleophilic addition of the phosphine, initially two double bonds should be in pyridine ring.)
Response:. The related pyridine rings are modified in Scheme 4 in the revised manuscript.
Reviewer 3 Report
In my opinion, this is a very well-performed manuscript from a theoretical physical organic chemistry point of view. The used Quantum Chemistry is state of the art for the task. The tools used to explain the reactivity and orbital interactions are excellent.
The studied mechanism is very complex. Nevertheless, the reaction profiles are convincing and probable. It is very welcome that the counterion is included, which in my opinion is much more critical than it is believed in standard Organic Chemistry.
It positively impresses me the use of the usually ignored, correction proposed by S.W. Benson for liquid phase entropy loss. Regarding that, I have some comments.
The exact value at 298.15 K is 2.55 kcal/mol. Sidney Benson has some typos in his Book because the numbers do not correspond with the formula. As far as I know, the correct numbers were proposed by Okuno (Okuno, Y. Chem.-Eur. J. 1997, 3, 212). The authors’ citation is similar to the work of Schoenebeck & Houk (reference 50 in the manuscript), but the article of Okuno precedes cited references. I have used it since 2006.
The section “Materials and methods” that I would name computational methods or methodology is too concise. It is unclear whether geometry optimizations were performed in solution or the continuum model was used only in single-point calculations. It must be clarified. I have one suggestion for future works depending on which is the case. I recommend geometry optimization in solution rather than only energy calculations using thermodynamic cycles to obtain Gibbs energy in solution. However, in such a case, SMD is the best choice instead of IEFPCM. Details about the last can be found in: R. F. Ribeiro, A. V. Marenich, C J. Cramer, D. G. Truhlar; J. Phys. Chem. B 2011, 115, 14556–14562.
It is also unclear if the Gibbs energies are taken as in the output of Gaussian 16, i.e., Standard State 1 atm, or corrected to standard state 1 M as corresponds to reaction in solution or for kinetic calculations, even in the gas phase. For each process with a change in the number of moles equal to -1 the correction would be -1.89 kcal/mol. Accordingly, it will be equivalent to 1.89 for a change of number of moles equal 1. It is essential to mention if it was done or to do if it was missed. For easiness, I use the sum of Benson’s correction plus the standard state change correction. I.e 4.44 kcal/mol. A detailed description of the use of the two corrections can be found in J. R. Alvarez-Idaboy and L. Reyes: J. Org. Chem. 2007, 72, 6580, which is only for a consult if necessary and does not need to be cited.
The free energy barrier for each reaction step is shown in the reaction profiles. It is very welcome; however, the double delta is incorrect. As in any reaction profile relative to reactants, variation in G is shown. The double delta is usually used to denote the free energy of solvation.
I strongly recommend publication after these minor points have been addressed. I apologize for any possible misinterpretations.
Author Response
Many thanks for your good suggestions. We revised the manuscript and improved its quality. Our changes and responses are listed below:
1 Comment (The exact value at 298.15 K is 2.55 kcal/mol. Sidney Benson has some typos in his Book because the numbers do not correspond with the formula. As far as I know, the correct numbers were proposed by Okuno (Okuno, Y. Chem.-Eur. J. 1997, 3, 212). The authors’ citation is similar to the work of Schoenebeck & Houk (reference 50 in the manuscript), but the article of Okuno precedes cited references. I have used it since 2006.)
Response:. The correct number 2.6 is changed to 2.55 and the paper (Okuno, Y. Chem.-Eur. J. 1997, 3, 212) is cited as reference 49 in the revised manuscript.
2 Comment (The section “Materials and methods” that I would name computational methods or methodology is too concise. It is unclear whether geometry optimizations were performed in solution or the continuum model was used only in single-point calculations. It must be clarified. I have one suggestion for future works depending on which is the case. I recommend geometry optimization in solution rather than only energy calculations using thermodynamic cycles to obtain Gibbs energy in solution. However, in such a case, SMD is the best choice instead of IEFPCM. Details about the last can be found in: R. F. Ribeiro, A. V. Marenich, C J. Cramer, D. G. Truhlar; J. Phys. Chem. B 2011, 115, 14556–14562.)
Response:. The section “Materials and methods” is changed to “Computational Methods”. Both Geometry optimizations and single-point calculations are performed in solution and are clarified in Computational Methods.
3 Comment (It is also unclear if the Gibbs energies are taken as in the output of Gaussian 16, i.e., Standard State 1 atm, or corrected to standard state 1 M as corresponds to reaction in solution or for kinetic calculations, even in the gas phase. For each process with a change in the number of moles equal to -1 the correction would be -1.89 kcal/mol. Accordingly, it will be equivalent to 1.89 for a change of number of moles equal 1. It is essential to mention if it was done or to do if it was missed. For easiness, I use the sum of Benson’s correction plus the standard state change correction. I.e 4.44 kcal/mol. A detailed description of the use of the two corrections can be found in J. R. Alvarez-Idaboy and L. Reyes: J. Org. Chem. 2007, 72, 6580, which is only for a consult if necessary and does not need to be cited.)
Response:. The Gibbs energies are corrected to standard state 1 M. The free energies in our work are recalculated based on the correction (4.44 kcal/mol), and the reaction profiles are modified accordingly. The paper (J. R. Alvarez-Idaboy and L. Reyes: J. Org. Chem. 2007, 72, 6580) is cited as reference 54.
4 Comment (The free energy barrier for each reaction step is shown in the reaction profiles. It is very welcome; however, the double delta is incorrect. As in any reaction profile relative to reactants, variation in G is shown. The double delta is usually used to denote the free energy of solvation.)
Response:. Double delta is changed to one delta to denote the free energy barrier in reaction profiles.